# Investigating on the Pavement Performance of Multi-Source Solid Wastes by Cement and Fly Ash

**DOI:** 10.3390/ma16196556

**Published:** 2023-10-04

**Authors:** Long Shan, Hongbo Li, Jing Zhao, Xuanshuo Zhang, Xinrui Kang, Xing Gao, Zhiyao Zhou

**Affiliations:** 1Department of Civil and Hydraulic Engineering, School of Civil and Hydraulic Engineering, Wencui Campus, Ningxia University, Yinchuan 750021, China; 12022131152@stu.nxu.edu.cn (L.S.); z17341244772@163.com (J.Z.); zxsnikea@163.com (X.Z.); kxinrui@163.com (X.K.); 15735861107@163.com (X.G.); zzy409644428@163.com (Z.Z.); 2Engineering Research Center for Efficient Utilization of Water Resources in Modern Agriculture in Arid Regions, Yinchuan 750021, China; 3Ningxia Research Center of Technology on Water-Saving Irrigation and Water Resources Regulation, Yinchuan 750021, China

**Keywords:** multi-source solid wastes, mechanical strength, frost resistance, mathematical model, microscopic morphology

## Abstract

In order to advance the utilization rate of multi-source solid wastes in the Ningxia region of China, 16 groups of pavement base mixtures were designed with cement and fly ash (FA) as binders, steel slag (SS), silicon manganese slag (SMS), and recycled crushed stone (RCS) as composite aggregates. The evolution laws of mechanical and frost resistance properties of the mixture were investigated by unconfined compressive strength (UCS), indirect tensile strength (ITS), freeze–thaw (FT), and ultrasonic detection tests. Then, the strength formation mechanisms were revealed by microscopic characterization technology. The mathematical models between UCS-ITS, UCS-ultrasonic amplitude, FT cycles-UCS damage, and frost resistance coefficient-relative dynamic elastic modulus *E*_r_ were established. The results show that cement content and curing age exhibited a positive effect on the mechanical strength and frost resistance of the mixture. When the replacement rate of SS was 60%, the mechanical strength and frost resistance were preferable. The *R*^2^ of the strength relationship models constructed was greater than 0.9, indicating high fitting accuracy. With the extension of the curing age, the cementitious products such as C-S-H (hydrated calcium silicate) and AFt (ettringite) developed entirely, and they were interlocked and cemented with each other, resulting in the micro-morphology developed from the three-dimensional network structure to the dense system. The macroscopic behavior incarnated that the mechanical strength and frost resistance of the mixture were significantly enhanced.

## 1. Introduction

With the continuous improvement of urbanization construction, the vigorous development of the iron and steel industry and coal-fired power generation in China, substantial multi-source solid wastes (MSSWs) are engendered, such as recycled crushed stone (RCS), steel slag (SS), silicon manganese slag (SMS), and fly ash (FA). By 2022, the comprehensive utilization rate of RCS was less than 12%, and the comprehensive utilization rate of industrial solid wastes was only 16% in the Ningxia Autonomous Region of China, which was not by a large difference from developed regions such as North America and Europe. The investigations demonstrated that forging 1 t of steel will form about 0.13 t of steel slag [1], making 1 t of manganese will produce about 10 t of silico-manganese slag [2,3], generating per trillion kW·h of power will emerge about 0.7 × 10^9^ t fly ash [4,5]. As the principal energy development province in China, Ningxia has a critical accumulation of MSSW. Under the leaching of precipitation, it significantly attacks the local ecological environment, land resources, and felicitous life of people [6,7,8]. The contaminant cycle process of the MSSW is introduced in Figure 1. In order to accomplish the target of high-quality development of material–environment–economy, prominent scholars at home and abroad have carried out the research wave of manufacturing concrete with high performance, low carbon emissions [9,10], and hoisting the utilization rate of MSSW.

At present, FA has been extensively applied in goaf, roadbed backfilling, and other fields because there are no rigorous standard for the particle size and chemical compositions [11,12,13]. However, regional differences trigger poor utilization of FA in extracting rare elements and glass preparation [14,15,16,17]. Mathapath et al. [18] investigated the effects of FA content and curing temperature on RCS and recycled asphalt pavement base or subbase materials. The results summarized that the mechanical properties of the mixture were preferable when the curing temperature was 40 °C, and the FA content was 15%. Xue et al. [19] analyzed the fatigue performance of RCS stabilized by FA and silica fume (referred to as “lime-fly ash”) by vibration compaction method and established relevant mathematical models, which demonstrated that the ITS of the mixture increased with the increase in lime-fly ash content. Thi et al. [20] claimed that a pavement base mixture stabilized by FA and cement can be manufactured by adding 10%–20% powder of FA in accordance with the national mechanical property standard.

The mineral compositions of SS and SMS are approximate to cement, and there is a small quantity of free calcium oxide (f-CaO) and magnesium oxide (f-MgO), indicating they possess micro-expansion characteristics and cementitious properties [21,22]. Therefore, SS and SMS can potentially be employed as high-quality pavement base materials. Pasetto et al. [23] constructed the preferable mixing ratio of SS-RCS aggregates in road-base mixture. Miah et al. [24] found that the alkaline system formed after the hydration of SS dedicated a high-quality environment for depolymerising vitreous in FA and stimulated the activity of C_2_S and C_3_S mineral phases in SS. Gao et al. [25] observed that the compactness of the SS surface cracks was better than natural aggregate, and the surface smoothness and compactness of the mixture was optimized after adding SS. Zhao et al. [26] investigated the expansion characteristics of SS and FA and found that the shrinkage value of SS mixture was smaller compared with FA. Aiban et al. [27] conducted a classification study on SS and comprehensively evaluated the relationship between SS content and road grade through laboratory tests and field applications. Frías et al. [28,29,30] incorporated 5% and 15% SMS micro powder into the mortar, which the strength of 7 days was lower than the reference group, but the strength in the 28 days and 90 days were adjacent to the reference group. At the same time, SMS powder enhanced the corrosion resistance of cement paste in Na_2_SO_4_ and NaCl media. Tamayo et al. [31] found that SMS as a building material met the leaching and geometric requirements of the current regulations. However, SMS will cause mechanical losses in the mixture, which was why it can only be applied in non-structural concrete.

The RCS can be crushed and screened to manufacture pavement base materials, alleviating the ecological tension caused by the natural sand and gravel shortage. At the same time, it also aligns with the direction of national green development. Trottier et al. [32] used the equivalent volume method, direct replacement method, and particle volume model to explore the frost resistance of RCS materials. Tang et al. [33] demonstrated that although the mechanical properties and durability of traditional gravel stabilized by cement and RCS stabilized by cement could not reach traditional gravel-RCS stabilized by cement, both answered low-grade road requirements. Li et al. [34] found that RCS lowered the strength of the mixture, but when the cement content was 4%, the strength in 7 days met the specification requirements of the specification. Zhang et al. [35] designed three ratios of RCS to clay brick (1:9, 5:5, 3:7), revealing that when the cement content was 4%, and the ratio of RCS to clay brick did not exceed 3:7, the shrinkage performance of the mixture answered the requirements of highway and first-class highway grassroots construction.

In summary, more researchers have adopted FA, SS, or RCS as pavement base materials, contributing theoretical support and technical guidance for upgrading the comprehensive utilization of MSSW. However, the utilization scale of solid wastes by incorporating SS, SMS, FA, and RCS to synthesize the pavement base mixture is still extremely deficient. At the same time, the comprehensive analysis of road performance and microscopic mechanisms of MSSW stabilized by cement and fly ash are less concerned. Therefore, cement and FA were regarded as the binder, and SS, SMS, and RCS were used as composite aggregates in this study. The unconfined compressive strength (UCS) test, indirect tensile strength (ITS) test, freeze–thaw (FT) test, and ultrasonic test were launched to analyze the influence of SS on the mechanical strength of the mixture. The internal structure and strength mechanism revealed by the hydration products and micromorphology were discussed in detail with the optimum ratio as the test object. The research results are conducive to stimulating the large-scale usage of MSSW in pavement engineering and elevating the utilization rate of MSSW.

## 2. Test Raw Materials and Methods

### 2.1. Raw Materials

SS was generated from the converter steel slag of the Zhizhong Industrial Solid Waste Disposal Company, Yinchuan, Ningxia, China. The volume stability by the test was qualified, the density was 3.15 g·cm^−3^, the water absorption rate was 1.86%, and the crushing value was 18.6%. SMS was supplied by SMS Factory in Ningxia, China, with a water absorption of 2.54% and density of 3.76 g·cm^−3^. RCS was obtained from Yinchuan, Ningxia, China, with a density of 3.04 g·cm^−3^, water absorption of 1.98%, and crushing value of 21.7%. When measuring the water absorption of SS, SMS, and RCS, the soaking time is the same for 24 h [36]. The cement was ordinary Portland P.O 42.5. FA was derived from the Xixia Thermal Power Plant in Yinchuan, Ningxia, China, with a specific surface area of 372.6 m^2^·kg^−1^. The particle size distribution of cement and FA was acquired by laser particle size detector (the detection range: 0.017–2000 μm, Mastersizer30000303081002, Malvern Company, Malvern, UK), which is displayed in Figure 2. An X-ray fluorescence analyzer (the analyzable elements: 11Na to 92U, S2RANGER LE03040428, AXS, Bruker, Karlsruhe, Germany) was conducted to detect the chemical compositions of raw materials, and the results are listed in Table 1. The main performance indexes of cement are summarized in Table 2.

Table 1 illustrates that the chemical compositions of raw materials are mainly SiO_2_, Al_2_O_3_, Fe_2_O_3_, CaO, and MgO, and the basic oxides account for a relatively high proportion, reaching more than 85% of the total mass, with potential pozzolanic activity and cementitious activity [37]. The micro-morphology of SS and SMS was observed by a scanning electron microscope (Magnification: 5×–1,000,000×, EVO 18, Zeiss GMBH, Oberkochen, Germany), and the results are summarized in Figure 3. The white irregular vitreous body is the RO phase, which belongs to the solid solution composed of div alent metal oxides. In addition, SS and SMS have cementitious properties and can enhance the compactness of the mixture due to the presence of C_2_S (dicalcium silicate) and C_3_S (tricalcium silicate). X-ray diffractometer (The rotation range of the goniometer: −10° to 168°, D8 ADVANCE, AXS, Bruker, Karlsruhe, Germany) was used to detect the mineral composition of raw materials (the samples were dried in a vacuum oven at 60 °C, and the powder samples after mechanical test were investigated by XRD), and the results are displayed in Figure 4. There are many dispersion peaks in the XRD pattern, indicating more amorphous glass phases.

### 2.2. Mixture Proportion

According to the gradation range of the standard (JTG/TF20-2015) [38], the median value of gradation was applied as the synthetic gradation of SS, SMS, and RCS. The comprehensive screening results of SS, SMS and RCS are exhibited in Figure 5. In the materials stabilized by cement and FA, the cement content of greater than 7% will not be conducive to economic benefits, and the cement content of less than 3% will be challenging to meet the requirements of road performance, demonstrating the cement content should be in the range of 3–7%. Moreover, Naceri et al. [39] and Thi et al. [20] have manifested that when the ratio of cement to FA was maintained between 1:3 and 1:5 and the FA content was between 10% and 20%, the pavement base mixture had superior mechanical properties. When the SMS content is 20%, the mechanical properties of the admixture in 7 days are equivalent to the reference group and the mechanical properties in 28 days are better than the reference group [40]. Combined with the pre-test, engineering experience, and specification requirements, the total content of cement and FA in the designed mixture was 20%, and the cement contents were 2%, 3%, 4%, and 5%, respectively. Due to the different places of production and environment, SS of 2.36–9.5 mm and 9.5–16 mm were used to substitute the RCS of the corresponding particle size in the test. According to the total proportion of 100% in the mixture, the mass proportion of SS and RCS can be obtained by conversion under different SS replacement rates (0%, 30%, 60%, and 90%), and the total proportion of both in the mixture was 41.6%. The replacement rates of SS were 0, 30%, 60%, and 90%, respectively. In addition, SMS and RCS were adopted as the composite aggregate of 0–2.36 mm and 16–26.5 mm particle size, respectively, accounting for 19.2% of each part. The ratio of mixtures designed are illustrated in Table 3.

### 2.3. Methods

#### 2.3.1. Compaction Test and Preparation of Specimens

According to the standard (JTGE51-2009) [41], the compaction test was performed to obtain the optimum moisture content (OMC) and maximum dry density (MDD) of 16 groups mixture. The compaction test was carried out by a multi-function automatic control electric compaction device (Degree of compaction: 98%, hammer times 60 ± 5 times·min^−1^, compaction preset number 0–999 times, JZ-2D, Tianyun instrument equipment company, Tianyun, China). Five water contents were set for each proportion, and the test was repeated twice, and the results were averaged, which was equivalent to ten samples for each proportion. Then, the cylindrical specimens with Φ 150 mm × 150 mm were prepared, which were utilized for the UCS, ITS, and FT tests. The compaction test and specimen preparation process are displayed in Figure 6.

#### 2.3.2. UCS and ITS Tests

The UCS and ITS tests were carried out on the specimens at 7, 28, 56, and 90 days of curing ages according to the standard (JTGE51-2009). The tests were conducted on the microcomputer-controlled electro-hydraulic servo universal testing machine (The loading speed: 1 mm·min^−1^, the maximum load pressure: 1000 kN, SHT 4106, Shanghai New think twice development company, Shanghai, China). The number of specimens in each ratio was more than 13, and the average value of test results were calculated. The UCS and ITS tests are exhibited in Figure 7.

#### 2.3.3. FT Test

According to the standard (JTGE51-2009), the FT test was launched on 7 groups of mixtures (2–0–0, 2–0, 2–30, 2–60, 2–90, 3–60, and 4–60), in which the aggregate in specimen 2–0–0 was only RCS. Each mixture set should prepare 18 Φ150 mm × 150 mm standard cylinder specimens, of which 9 were freeze–thaw specimens, and 9 were parallel specimens to measure the UCS of pre-freeze–thaw specimens. Firstly, the freeze–thaw specimens that were cured for 28 days (after 27 days of curing, the specimens were immersed in water for 1 day, and the water surface was 2.5 cm higher than the top surface of the specimen) were settled in a −18 °C low-temperature chamber (Rated power 320 W, 728, Qingdao Tao Qiang trading company, Qingdao, China) and maintained at a constant temperature for 16 h. In addition, we should guarantee that each test block can contact the same freeze–thaw environment. When the 20 mm pores were kept between the test blocks, the full circulation of cold air was ensured. Otherwise, each test block will produce different frost heave forces due to contacting with different environments, which will lead to different pore expansion degrees and inconsistent damage forms, resulting in a large of errors in the test results and becoming an invalid test. Then, the specimens were placed in a 20 °C tank and maintained at an invariant temperature for 8 h. Finally, the specimens were removed and put on a horizontal dry desktop. After standing for 15 min, the specimens were weighed and their average value was taken: a complete freeze–thaw process. The UCS and mass of the specimens were recorded after 0, 5, 10, 15, 20, 25, 30, 35, and 40 freeze–thaw cycles. At the same time, the freeze-resistance of the mixtures was evaluated comprehensively from the appearance, mass, and UCS damage changes, respectively. The damage amount of UCS is calculated according to Formulas (1) and (2), and the FT test is manifested in Figure 8.
(1)$$D_{\mathrm{UCS}}=1-BDR$$
(2)$$BDR=\frac{R_{\mathrm{dc}}}{R_{c}}$$
where *D*_UCS_ is the damage amounts of UCS (%), *BDR* is the coefficient of frost resistivity (%), *R*_dc_ is the UCS of specimens after n freeze–thaw cycles (MPa), and the *R*_c_ is the UCS of non-freeze–thaw specimens (MPa).

#### 2.3.4. Ultrasonic Test

The relative dynamic elastic modulus *E*_r_ can describe the degree of damage inside the mixture [42,43]. Based on the principle that ultrasonic waves will form reflections and refractions at different media interfaces, it was performed using a non-metal ultrasonic detector (acoustic time accuracy: ±0.05 μs, ZBL-U520, Beijing Zhibolian technology company, Beijing, China) to investigate the relationship between UCS-ultrasonic amplitude and BDR-*E*_r_ of MSSW stabilized by cement and FA. Frist of all, under the premise of ensuring the accuracy of sound time to be ±0.05 μ, the sound time was reduced to zero, and the Vaseline coupling agent was applied to the transmitting and receiving monitor. Then, the two monitors were attached to the upper and lower circular measurement areas of the mixture to guarantee that the detector could smoothly transmit and receive signals. Finally, five points were taken from each survey area, and the average value was calculated [37,44,45]. *E*_r_ is calculated by Equation (3), and the ultrasonic test is showcased in Figure 9.
(3)$$Er(n)=\frac{E_{n}}{E_{0}}=\frac{V_{n}^{2}}{V_{0}^{2}}=\frac{{\left(l/t_{n}\right)}^{2}}{{\left(l/t_{0}\right)}^{2}}=\frac{t_{0}^{2}}{t_{n}^{2}}$$
where *E*_r_ is the relative dynamic elastic modulus of specimens, *E*_n_ is the dynamic elastic modulus of specimens after n freeze–thaw cycles (MPa), *E*_0_ is the relative dynamic elastic modulus of specimens before freezing and thawing (MPa), *V*_n_ is the longitudinal wave velocity of specimens after n freeze–thaw cycles, *V*_0_ is the longitudinal wave velocity of specimens before freezing and thawing (km·s^−1^), *L* is the length of specimens before freezing and thawing (mm), *T*_n_ is the sound time of the specimens after n freeze–thaw cycles, and *T*_0_ is the sound time of the specimens before freezing and thawing.

#### 2.3.5. Microscopic Test

The mineral compositions and microstructure of specimen 2–60 at 7, 28, 56, and 90 days were analyzed by XRD and SEM microscopic detection techniques to fully understand the hydration reaction mechanisms of MSSW stabilized by cement and FA. Firstly, the central binder was taken as microscopic test samples after the UCS or ITS test and settled in anhydrous ethanol to terminate hydration. Then, the samples were dried in the vacuum oven at 60 °C before detection. Eventually, the powder samples were examined by XRD, and the flat samples were checked by SEM.

## 3. Test Results and Analysis

### 3.1. Compaction Test

Figure 10 summarizes the compaction test results of 16 groups mixtures. It can be seen that there was a negative correlation between the SS replacement rate and the OMC of the mixtures. Taking 2% cement content as an example, when the SS replacement rate increased by 30%, the average OMC of the specimens decreased by 3.8%, and the corresponding average MDD increased by 5.3%. The reason is that SS has a smoother surface, more negligible water absorption, and greater density than RCS. The smoother surface and more negligible water absorption of SS reduced the OMC of the specimens, while the greater density increased the MDD of the specimens. With the increase in cement content, the OMC of the specimen increased, mainly because the specific surface area of cement was larger than FA. Therefore, the water absorption performance of the specimen was enhanced with the increase in cement content, which led to the rise of OMC. At the same time, the cement content was positively correlated with the proportion of gelling substance in the mixture, and the gelling substance had a strong ability to combine with water [46], leading to an increase in the water requirement of the mixture to a certain extent.

### 3.2. UCS Test

The UCS test results of the specimens 2–0 to 2–90 at different ages are exhibited in Figure 11, which reveals that the UCS of the mixtures is positively correlated with the curing age. Nevertheless, the replacement rate of SS is positively correlated with the strength and then negatively correlated with the strength. At the 7 days of curing age, the UCS of specimens 2–30, 2–60, and 2–90 were 3.68, 4.06, and 3.89 MPa, which were 6.1%, 17.0%, and 11.5% higher than specimen 2–0, respectively, indicating 60% SS replacement had the most apparent promptitude in UCS of the mixtures. This can be attributed to two aspects. On the one hand, the crushing value of SS is smaller, and the surface texture is rougher than RCS, which gives SS a stronger ability to resist destruction than RCS. On the other hand, the C_2_S, C_3_S and RO phase in SS participated in the hydration reaction to generate more C-S-H and other cementitious products, strengthening the degree of compaction of the mixtures. The UCS of specimen 2–90 was 4.06% lower than specimen 2–60, demonstrating that too much SS led to the superposition of its micro-expansion effect, and the compactness of the mixture was depressed to a certain extent, which was not conducive to the development of the strength of the mixture. The UCS growth rate of specimen 2–60 increased with the increase in curing age, but the increment decreased gradually. At the 28 days, the growth rate of specimen 2–90 was the highest, exceeding 85%. The principal reason was that in the early stage of hydration, active substances such as SiO_2_ and CaO in SS and SMS participated in the hydration reaction in large quantities, forming more cementitious products, such as C-S-H, C-A-H, C-A-S-H, and AFt, resulting in the interfacial transition zone (ITZ) between the raw materials becoming more compact, which directly made the UCS growth rate of specimen 2–90 the largest at 7 days. This also led to the gradual reduction in the proportion of active substances in the mixture system, so the UCS growth rate of the mixtures decreased in the later stage of hydration (56 days and 90 days). In summary, the UCS of the mixtures was superior when the SS replacement rate was 60% under the 2% content of cement.

Based on the superior SS replacement rate of 60%, the UCS development laws of mixtures with different cement contents (2%, 3%, 4%, and 5%) at 7, 28, 56, and 90 days were explored, as shown in Figure 12. It can be observed that the UCS of the mixtures rose with the increase in cement content and curing age. At 7 days, the UCS of specimens 3–60 and 4–60 were significantly higher than specimen 2–60, which increased by 1.12 MPa and 1.59 MPa, respectively. This is because with the addition of cement content, the proportion of CaO in the mixture increased, and more Ca(OH)_2_ was generated, which stimulated the early-aged hydration reaction of the mixture. When the cement content was 5%, the proportion of CaO in the mixture continued to increase, but the UCS of specimen 5–60 was a mere 0.77 MPa higher than specimen 4–60. This occurred because excessive cement content inhibited the secondary hydration of active substances such as Al_2_O_3_ and SiO_2_. Therefore, when the mixture has higher early strength requirements in engineering applications, measures such as optimizing the aggregate gradation and optimize construction technology should be seriously considered. The average UCS growth rate of specimens with different cement content reached 85.4% at 28 days, mainly depending on the micro-aggregate effect and secondary hydration reaction. At 56 days and 90 days, the average growth rate of UCS of the specimens decreased significantly, which was caused by the slow reaction rate in the later stage due to the completion of hydration of the mixed liquid in the earlier stage.

The mathematical models between UCS and cement content under the superior SS substitution 60% were acquired by fitting to investigate the UCS variations of mixtures with different curing ages. These formulas were designed to qualitatively show the relationship between cement content and UCS of mixtures under different curing ages, hoping to predict by fitting the formula models (linear function or exponential function). The fitting results of mathematical models are displayed in Figure 13. The correlation coefficient *R*^2^ of the fitting curves exceeded 0.97, manifesting that the models were in good agreement with the experimental results. Moreover, the curves can accurately reflect the relationship between UCS and cement content at different curing ages under 60% SS substitution rate, which contributed to the theoretical basis for practical engineering applications.

Figure 14 more intuitively shows the influence of the interaction effect of cement content and SS substitution rate on the UCS of the mixtures at different ages. It can be observed that it exhibits similar rules at 7, 28, 56, and 90 days. So only the UCS of the mixtures at 7 days are analyzed here. As you can see, when the cement content was 5%, the SS substitution rate corresponding to the maximum UCS of the mixtures was between 60% and 90%, reaching 8.16 MPa, which indicated that the cement content and SS substitution rate had the most substantial interaction effect on the UCS of the mixture. Therefore, based on the 60% SS replacement rate, the UCS will still increase by continuing to incorporate the appropriate amount of SS, which provides a reference value for optimizing the scheme in actual projects.

### 3.3. ITS Test

ITS is a critical index of the capability of the mixture to resist tensile failure. Figure 15 summarizes the ITS test results of the mixtures, showing the same regulations in different cement content. So only the ITS of 2% cement content is analyzed here. Similarly, ITS reached the maximum when the SS substitution rate was 60%, which presented consistent with the UCS results. Moreover, the average ITS growth rates of the mixtures at 28, 56, and 90 days were 44.3%, 22.3%, and 14.4%, respectively, showing a decreasing trend. This is because with the continuous extension of curing age, the proportion of active substances involved in hydration in the system gradually decreases, and the strength increases less in the later stage.

### 3.4. Fracture Patterns in UCS and ITS Tests

Figure 16 describes the fracture patterns of the specimens in UCS and ITS tests and shows that the fracture patterns of both are distinct. In the UCS test, cracks first occurred in the ITZ and gradually expanded to the binder. In the process of pressurization, tiny cracks appeared in the specimen and accumulated continuously, and the width became larger and then several penetrating prominent cracks were formed, finally leading to the failure of the specimen. In the ITS test, the cracks extended up and down both sides after forming in the ITZ and deflected when crossing to the relatively weak side of the strength. Eventually, the failure occurred when a penetrating main crack was generated on the bottom surface of the specimen. In addition, the evenly distributed SS, SMS, RCS, cement, and FA binder and the hydration products in the mixture can be explicitly observed from the ITS section failure diagram.

### 3.5. UCS and ITS Analysis of Fitting Results

According to the UCS and ITS test results, there was a positive correlation between the UCS and ITS of the mixtures. Based on this, the relationship between UCS and ITS of mixtures was qualitatively analyzed by power, linear, exponential, and polynomial functions. The results are concluded in Figure 17. It can be summarized that the correlation coefficients of the four fitting models were higher than 0.915, which could accurately reflect the relationship between UCS and ITS of the mixtures. Therefore, when the ITS of the mixtures was not easily accessible, the mathematical model of the power function between the UCS and ITS can be applied to estimating the ITS of the mixture.

### 3.6. FT Test

#### 3.6.1. Appearance Analysis of the Mixtures

Frost resistance is an essential index to evaluate the durability of pavement base mixture. Ningxia is a typical seasonal freezing area, and the pavement base material is vulnerable to freeze–thaw damage, which will weaken its mechanical strength and durability. Therefore, carrying out the frost resistance test of MSSW stabilized by cement and FA is particularly critical.

Figure 18 showcases the appearance of the specimens at 0, 10, 20, and 40 cycles of freeze–thaw. It can be observed that the appearance of the specimens before freezing and thawing had local unevenness, which was mainly caused by uneven stirring or uneven pressing when preparing the specimen. In addition, specimen 3–60 had the best appearance integrity before freezing and thawing, which fully demonstrated that the SS replacement rate of 60% and the appropriate cement content can enhance the compactness of the mixture. After freezing and thawing for 10 cycles, there were some mesh cracks on the surface of specimens 2–0–0 and 2–0, but the appearance of other specimens was integrated. After 20 cycles of freezing and thawing, obvious cracks appeared on the surface of the specimens, and the fine aggregate began to fall off at different degrees, resulting in some exposure of SS and RCS. After 40 freeze–thaw cycles, the coarse aggregates in specimens 2–0–0, 2–0, and 2–90 became loose and began to fall off, as shown in Figure 18. However, when the SS replacement rate was 30% and 60%, the appearance damage of the specimens was relatively small, and only a tiny amount of fine aggregate fell off. In conclusion, the frost resistance of specimens 2–30, 2–60, and 3–60 was preferable.

#### 3.6.2. Quality Changes of the Mixtures

Figure 19 summarizes the quality of the specimens under different freeze–thaw cycles. It could be seen that the quality of all samples showed an increasing trend before 20 freeze–thawing cycles. During the freezing process, the water in the specimen turned into ice at low temperatures, and cracks appeared inside the specimen. During the melting process, the ice in the specimen melts into water at high temperatures, but the cracks formed by the last frost heave are still there, providing more room for the next frost heave. After multiple freeze–thaw cycles, the cracks accumulated, and the width became larger, resulting in severe water immersion. The quality peaks of specimens 2–0–0 and 2–0 appeared at 20 and 25 freeze–thaw cycles, respectively. This was because the incorporation of SMS stimulated the hydration reaction of the specimen, increasing its frost resistance. Different from specimens 2–0–0 and 2–0, the quality peak of specimen 2–90 appeared at 30 freeze–thaw cycles, and the quality decreased slowly after 30 freeze–thaw cycles. In addition, the quality after 40 freeze–thaw cycles was still more significant than before freezing and thawing. The reason was the micro-expansion effect of the SS was increased in the freeze–thaw process, causing the mixture to tighten gradually and the surface binder to fall off. However, the hydration of active substances such as C_2_S and C_3_S in SS enhanced the freezing resistance of the mixture, which explained why the quality of 2–90 was still higher than that before freezing and thawing after 40 cycles of freezing and thawing. At the same time, because the micro-expansion effect of SS in specimens 2–30 and 2–60 was not noticeable, the quality of the specimens exhibited a slow upward trend during the whole FT test. In addition, the mass growth rate of specimens 2–60, 3–60, and 4–60 decreased successively, indicating that the addition of cement strengthened the freezing resistance of specimens. The reason was that CaO, as the main component of cement, could produce more C-S-H and other gelling products in the hydration reaction, which filled the pores of the mixture and hindered the formation of cracks in the mixture.

#### 3.6.3. UCS Damage Amount of the Mixtures

Figure 20 summarizes the UCS damage amount of the specimens with the rise of freeze–thaw cycles. The mathematical models of the freeze–thaw cycles and UCS damage were acquired by linear regression analysis. The correlation coefficients *R*^2^ were greater than 0.97, illustrating that the freeze–thaw cycles of the specimen were in superior agreement with the UCS damage. It could be seen that when the freeze–thaw cycles continued to extend, the UCS damage gradually increased. After 40 freeze–thaw cycles, the UCS damage of specimens 2–0–0 and 2–0 were 66.6% and 60.9%, respectively, which were significantly greater than specimen 4–60 (15.4%), demonstrating that SS and SMS can dramatically alleviate the UCS damage of specimens. In addition, the fitting curves of UCS damage and freeze–thaw cycles of specimens 2–0–0 and 2–0 were the quadratic parabola, and the rest of specimens containing SS manifested the linear correlation. The principal reason is that the water absorption rate of RCS is greater than SS. Before 30 freeze–thaws, the cracks inside the mixture continued to expand, causing more free water to flow in and promoting greater frost heave forces. Therefore, the UCS damage of the early stage of the specimen ascended rapidly. After 30 freeze–thaw cycles, the internal porosity of the mixture decreased, resulting in the weakening of the active cooling pore water function of RCS, which made the UCS of the specimen exhibit a slow upward trend. Compared with specimen 2–0–0, the initial UCS damage of specimen 2–0 was less because active minerals such as C_2_S and C_3_S in SMS stimulated the hydration reaction more, thus enhancing the frost resistance of the specimen. At 8 freeze–thaw cycles, the fitting curves of UCS damage and freeze–thaw cycles of specimens 2–30 and 2–90 were crossed. This occurred because specimen 2–90 contained more SS. With the continuous extension of the freeze–thaw cycle, the decrease in compactness led to a higher rise in UCS damage than specimen 2–30. With the increase in cement content, the slope of the fitting curve of specimens 2–60, 3–60, and 4–60 decreased in turn. The reason was that the cementitious yield increased with the rise of cement content, which optimized the pore structure of ITZ and stimulate the mixture density.

### 3.7. Ultrasonic Test

According to the characteristic of forming refraction and reflection of ultrasonic waves in different ITZ, the mechanical strength inside the mixture can be analyzed. Li et al. [44,45] demonstrated that the ultrasonic amplitude was positively correlated with the UCS of the specimen, so the specimen at 7 days was measured by ultrasonic. At the same time, the measured sound wave was converted into a spectrogram [47] by the fast Fourier transform, and the amplitude value was recorded from the spectrogram. Then, the average amplitude of each specimen was calculated to investigate the linear regression model between UCS and the ultrasonic amplitude of the specimen, as shown in Figure 21.

It can be concluded from Figure 21 that the UCS of the specimen had a linear relationship with the ultrasonic amplitude. The correlation coefficient *R*^2^ of the fitting curve was 0.969, which attested that the mathematical model was in high agreement with the experimental value and could furnish theoretical support for the strength detection of road base in practical projects.

Zhao et al. [48] demonstrated that the *BDR* of the mixture was approximately the power function relationship with *E*_r_. Therefore, after the freeze–thaw test, the fitting analysis was carried out on the relationship between *BDR* and *E*_r_ of the specimen, and the results are displayed in Figure 22.

Figure 22 indicates that the *E*_r_ is positively correlated with the *BDR* of the specimen, and the relationship between both presents the power function. The correlation coefficient *R*^2^ was 0.946, which proved that the fitting results were in precise agreement with the experimental values. Therefore, ultrasonic testing technology can detect the UCS damage of the specimen, and the power function can be used to calculate the *BDR* of the mixture.

### 3.8. Microscopic Test

#### 3.8.1. Analysis of Mineral Compositions

Figure 23 presents the XRD test results of mineral compositions of the preferable SS replacement rate specimen 2–60 at different curing ages. It can be observed that the main mineral components were Ca(OH)_2_, C-S-H, C-A-H, C-A-S-H, C_2_S, C_3_S, Aft, and RO phase. At the 7 days, the characteristic peak value of Ca(OH)_2_ was the largest, followed by AFt, C_2_S, and C_3_S. This occurred because SS and cement contained magnanimous CaO, and some generated Ca(OH)_2_ by early hydration. The [SiO_4_] and [AlO_4_] vitreous structures in SS and SMS were gradually depolymerized, while Ca^2+^ and OH^−^ in the liquid phase composed the material basis for forming AFt. Moreover, AFt can stimulate the connection between hydration products and enhance the strength of the mixture, which was consistent with the results of UCS and ITS tests. In addition, due to the lower early activity of C_2_S and C_3_S, the hydration rate was slower, so the peak values of C_2_S and C_3_S were smaller. With the deepening of the hydration process (28 days), a large amount of CaO began to hydrate, resulting in a significant increase in the peak value of Ca(OH)_2_. Characteristic peaks of C_2_S and C_3_S decreased, and C-S-H, C-A-H and geopolymer C-A-S-H increased compared with the specimens of 7 days. The reason was that the raw materials provided sufficient Si and Al sources for the hydration system. At 56 days, the gelation ratio further increased, and the characteristic peak value of C_3_S almost disappeared. However, there was still a small amount of C_2_S. At 90 days, C-S-H, C-A-H, and C-A-S-H reached the peak, and the characteristic peaks of the C_2_S and C_3_S gradually disappeared. In summary, the primary source of the early strength of the mixture was the hydration of cement.

There was not much distinct in the peak value of RO phase because the RO phase was dominated by solid solution and had low activity, which mainly supplied the deposition platform for hydration products. Compared with the specimen of 28 days, the rise of Ca(OH)_2_ characteristic peak values at 56 days and 90 days was smaller. This is mainly attributed to two aspects. On the one hand, SS provided a sufficient Si source for the reaction system to react with Ca(OH)_2_ to form C-S-H gel. On the other hand, the strong alkalinity of Ca(OH)_2_ stimulated the secondary hydration of SS, SMS, and FA, which consumed a large amount of Ca(OH)_2_.

#### 3.8.2. Analysis of Micro Morphology

Figure 24 exhibits the SEM test results of the preferable SS replacement rate specimen 2–60 at different curing ages. It can be concluded that the formation of the macroscopic strength of the mixture was the process of the hydration product from the amorphous loose gel to the low and high crystallinity. At the 7 days, a small amount of scattered fibrous C-S-H, rod-shaped AFt, hexahedral C-A-H, and CH gel were observed to intertwine and form the spatial network structure containing many cracks and pores. At 28 days, the geopolymer C-A-S-H began to emerge, and more CH appeared. At the same time, the hydration products developed and expanded, and the number of cracks and pores decreased, indicating that the CaO in SS and cement began to hydrate to form CH and crystallize, which was consistent with the XRD results. At 56 days, the spatial structure changed from loose honeycomb to dense and compact. However, there were still a few of cracks and pores. C-S-H, C-A-H, C-A-S-H, CH, and AFt cementitious materials continued accumulating, among which were rod-shaped. At the same time, AFt played the role of supporting aggregates and stimulating the complete development of aggregates. The hydration products at 90 days were interwoven and stacked, and the microstructure manifested the most compact state, demonstrating that the hydration process was practically over. In addition, many FA particles were wrapped in the inner layer of the binder, as shown in Figure 24d. The reason was that SiO_2_ in FA and SMS was activated in the later stage of hydration and secondary hydration reaction, which occurred to form a dense combination. In conclusion, the mutual excitation of binder (cement and FA) and MSSW gradually optimized the micro-morphology of the mixture and ultimately reflected the improvement of macro-strength.

## 4. Conclusions

In this study, the UCS test, ITS test, FT test, ultrasonic damage detection, and microscopic characterization methods were utilized to investigate the mechanical and freeze–thaw behavior evolution regulations of MSSW stabilized by cement and FA and its microscopic hydration mechanisms were specially analyzed. The chief conclusions are as follows:(1)The UCS and ITS of the mixture exhibited a positive correlation with the cement content and curing age. However, the replacement rate of SS was positively correlated and then negatively correlated with the strength. At the 7 days, the UCS of specimens 2–30, 2–60, and 2–90 were 3.68, 4.06 MPa, and 3.89 MPa, respectively, which were 6.1%, 17.0%, and 11.5% higher than specimen 2–0, respectively. When the SS substitution rate was 60%, the mechanical properties of the mixture were preferable. With the increase in cement content, the UCS of the specimens manifested an upward trend, but the increase values of UCS gradually declined. The correlation coefficient *R*^2^ of the power function in the ITS-UCS relationship model was 0.937, indicating that the fitting was high. The *R*^2^ in the linear model of UCS-ultrasonic amplitude was 0.969, which was in good agreement with the experimental values.(2)The UCS damage of the mixture increased first and then decreased with the rise of the SS substitution rate. When the freeze–thaw cycle continued to extend, the appearance of the specimens had different degrees of damage, mainly manifesting as the detachment of fine aggregates and the weakening of the mosaic ability between coarse aggregates, so the strength was also weakened. However, when the SS substitution rate was 30% and 60%, the appearance of the specimens was relatively complete, and the damage amount of UCS was small. The *R*^2^ of the FT cycles-UCS damage and the *BDR*-*E*_r_ mathematical models were more than 0.945, which supplied the theoretical basis for practical engineering applications.(3)The results of XRD demonstrated that cementitious products such as C-S-H and AFt increased with the extension of curing age. The hydration of CaO in SS and SMS led to a substantial increase in the peak value of Ca(OH)_2_ of the mixture at 28 days. The RO phase mainly presented the deposition platform for hydration products. SEM characterization results indicated that the crystallinity of C-S-H and AFt in the mixture system increased dramatically. The microstructure eventually transformed into a dense association from a three-dimensional network structure with many holes and cracks.(4)The comprehensive test results showed that when the SS replacement rate was 60%, the mechanical strength and frost resistance of the mixture were preferable under the same cement content, which can be utilized as a novel environmental protection material and had great potential for based pavement mixture.

## 5. Discussion

The effects of SS replacement rates, cement dosages, and curing ages on the pavement performance of the mixture were investigated in this paper. As the semi-rigid base material, MSSW stabilized by cement-fly ash yet to be applied on a large scale in China and was only investigated in the laboratory. In order to comprehensively explore the road performance of the mixture and be involved in practical engineering, several aspects of research still demand to be further carried out, mainly as follows:(1)This paper exhibited the strength enhancement mechanisms of the mixture by microscopic characterization technologies. The next step should be schemed of the change regulations of pore size, pore shape, and porosity with the hydration process, and abundantly reveal the strength evolution mechanisms of the mixture.(2)This paper analyzed the compressive strength, splitting strength, and freezing resistance of the mixture. We should discover the evolution characteristics of the shear failure resistance, shrinkage resistance, and chloride ion penetration resistance of the mixture in the next step.(3)The paper has a shortage of evaluations on the environmental and economic benefits of the production stage of the mixture so that the next step can be the comprehensive assessment of the carbon emission and carbon reduction effect of the mixture by comparison with the traditional base materials.

## Figures and Tables

**Figure 1 materials-16-06556-f001:**
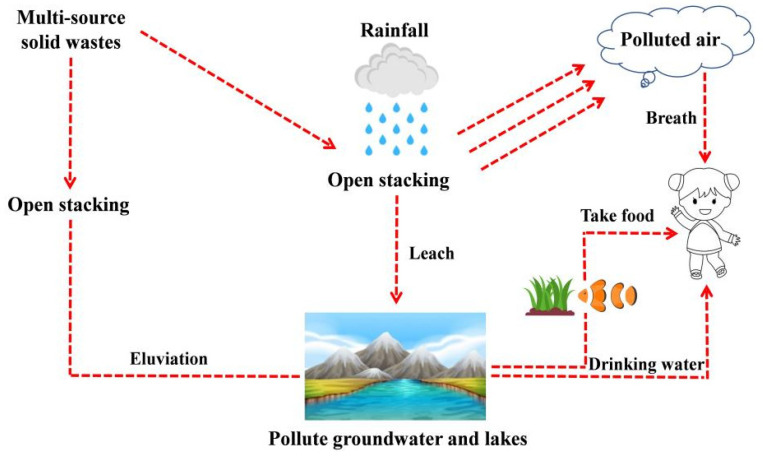
The contaminant cycle of multi-source solid wastes.

**Figure 2 materials-16-06556-f002:**
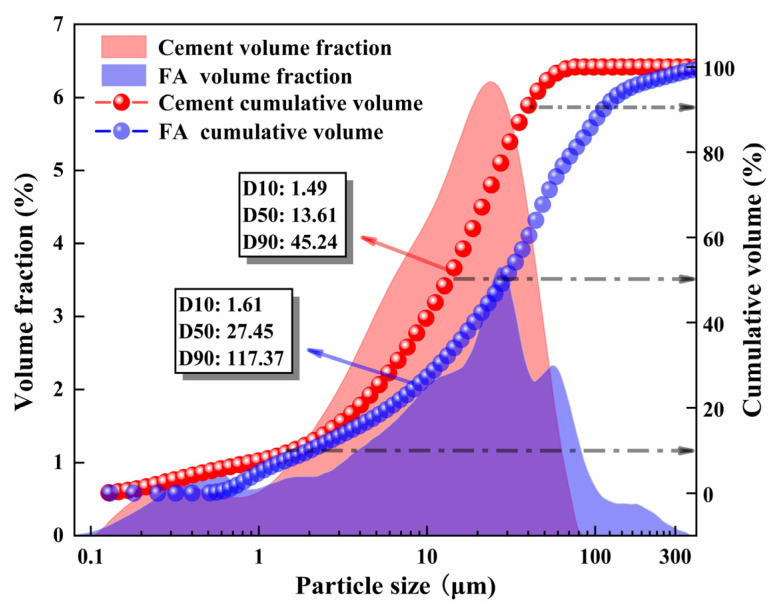
Particle size distribution of cement and FA.

**Figure 3 materials-16-06556-f003:**
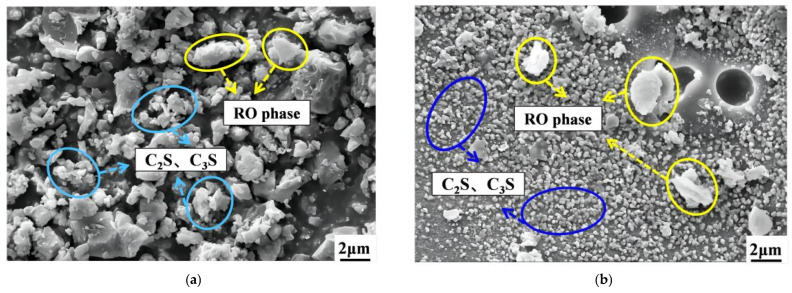
Micro morphology of SS and SMS: (**a**) SS and (**b**) SMS.

**Figure 4 materials-16-06556-f004:**
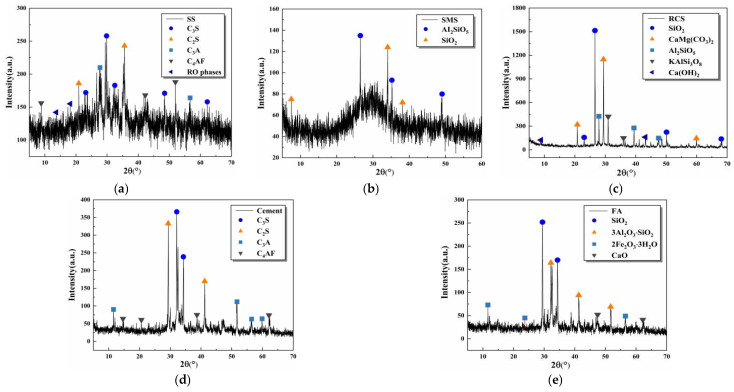
Mineral compositions of raw materials: (**a**) SS, (**b**) SMS, (**c**) RCS, (**d**) cement, (**e**) FA.

**Figure 5 materials-16-06556-f005:**
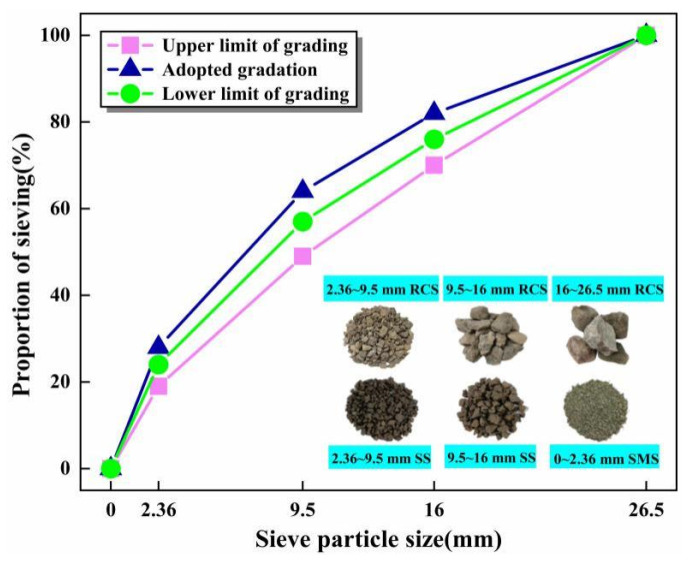
Sieve particle size curve.

**Figure 6 materials-16-06556-f006:**
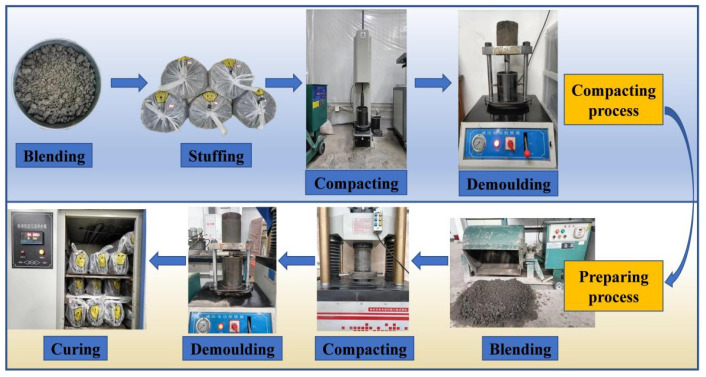
Compacting and preparing processes of the mixtures.

**Figure 7 materials-16-06556-f007:**
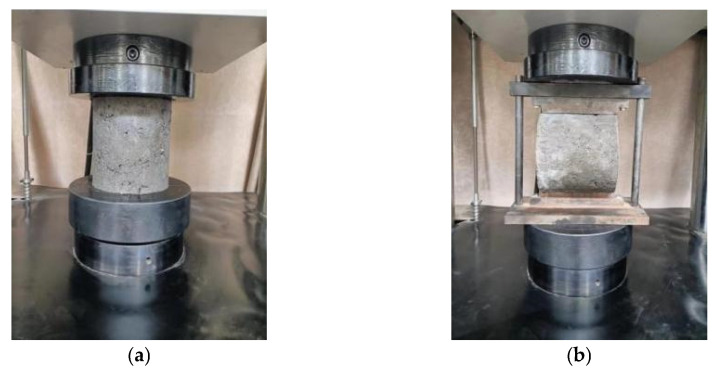
USC test and ITS test: (**a**) UCS test and (**b**) ITS test.

**Figure 8 materials-16-06556-f008:**
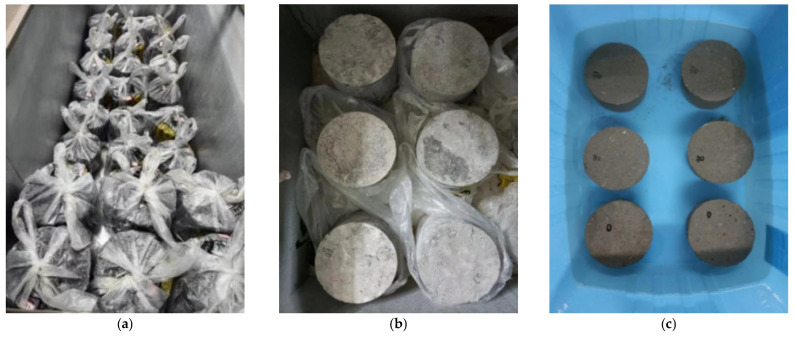
FT test: (**a**) bulk freezing test, (**b**) local freezing test, (**c**) thawing test.

**Figure 9 materials-16-06556-f009:**
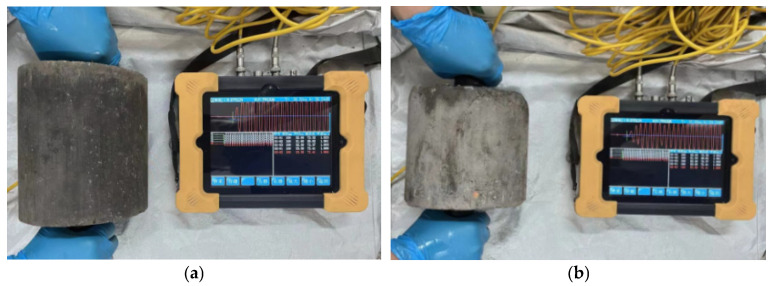
Ultrasonic test: (**a**) ultrasonic test in the UCS test and (**b**) ultrasonic test in the FT test.

**Figure 10 materials-16-06556-f010:**
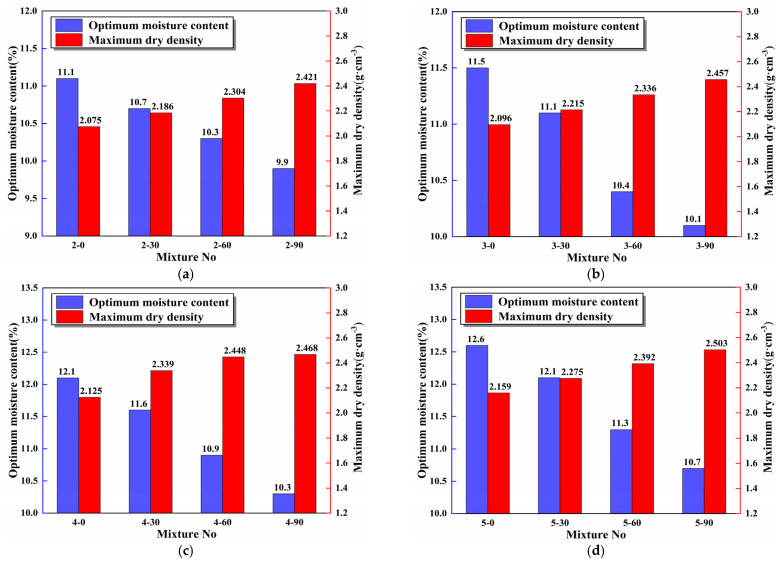
Test results of MDD and OMC of the mixtures of different cement contents: (**a**) 2% cement content, (**b**) 3% cement content, (**c**) 4% cement content, (**d**) 5% cement content.

**Figure 11 materials-16-06556-f011:**
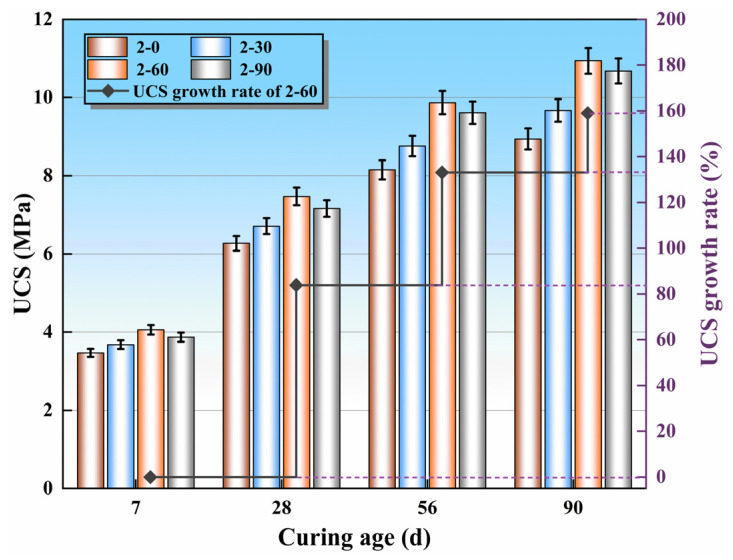
The UCS of the mixtures with various SS replacement rates.

**Figure 12 materials-16-06556-f012:**
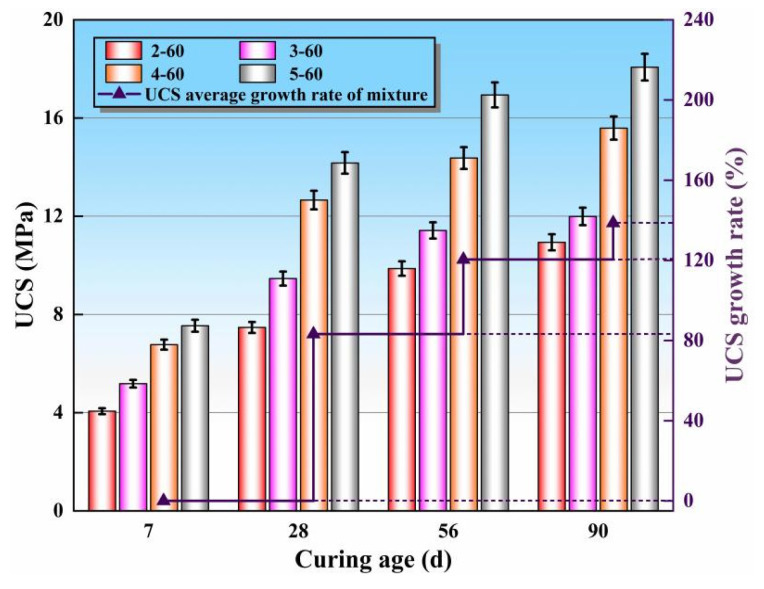
The UCS of the mixtures with various cement contents.

**Figure 13 materials-16-06556-f013:**
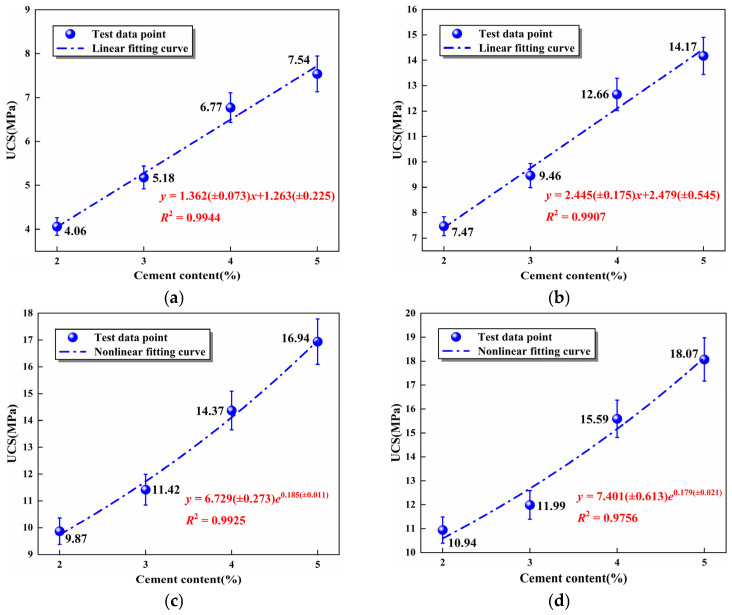
The relationship between UCS and cement content of the specimens at different curing ages: (**a**) 7 days, (**b**) 28 days, (**c**) 56 days, (**d**) 90 days.

**Figure 14 materials-16-06556-f014:**
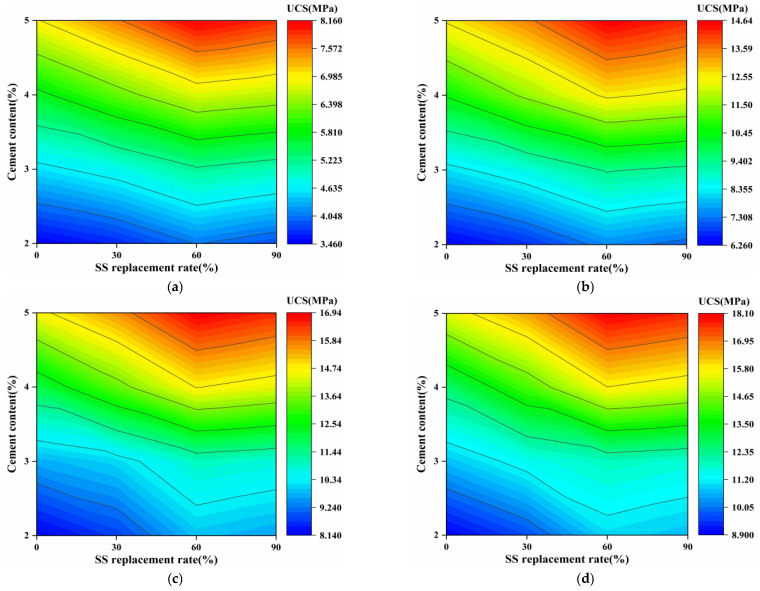
The UCS of the mixtures with various cement contents and SS replacement rates at different curing ages: (**a**) 7 days, (**b**) 28 days, (**c**) 56 days, (**d**) 90 days.

**Figure 15 materials-16-06556-f015:**
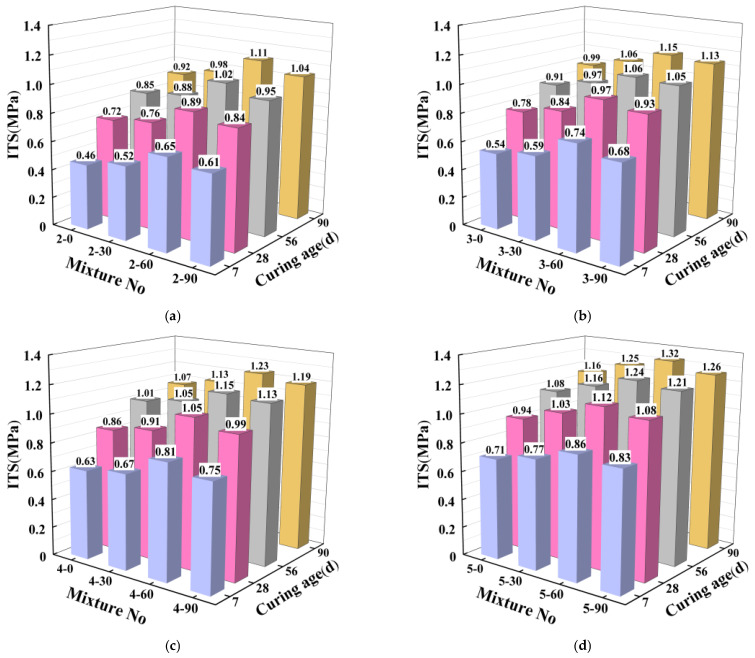
The ITS of the mixtures with various cement contents: (**a**) 2% cement content, (**b**) 3% cement content, (**c**) 4% cement content, (**d**) 5% cement content.

**Figure 16 materials-16-06556-f016:**
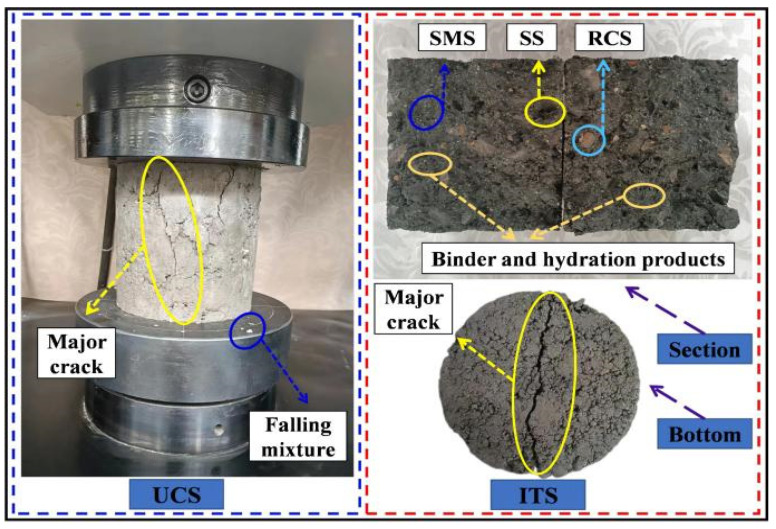
The fracture patterns of UCS and ITS tests.

**Figure 17 materials-16-06556-f017:**
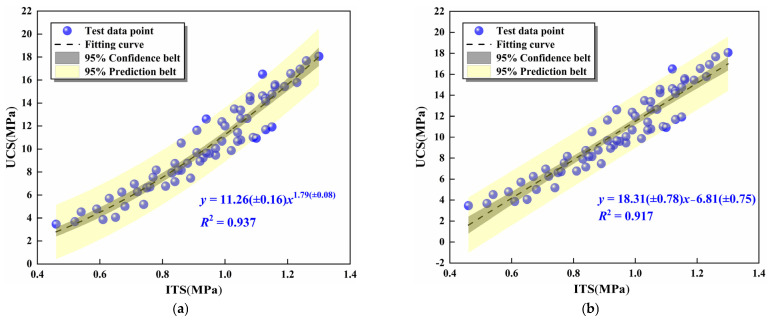
The mathematical models between UCS and ITS: (**a**) the power function, (**b**) the linear function, (**c**) the exponential function, (**d**) the polynomial function.

**Figure 18 materials-16-06556-f018:**
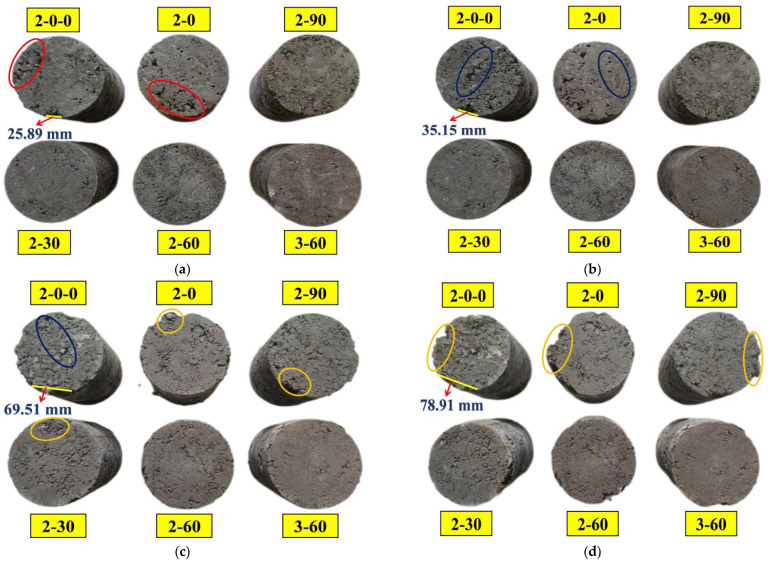
Appearance of the mixtures with various freeze–thaw cycles: (**a**) 0 cycles, (**b**) 10 cycles, (**c**) 20 cycles, (**d**) 40 cycles.

**Figure 19 materials-16-06556-f019:**
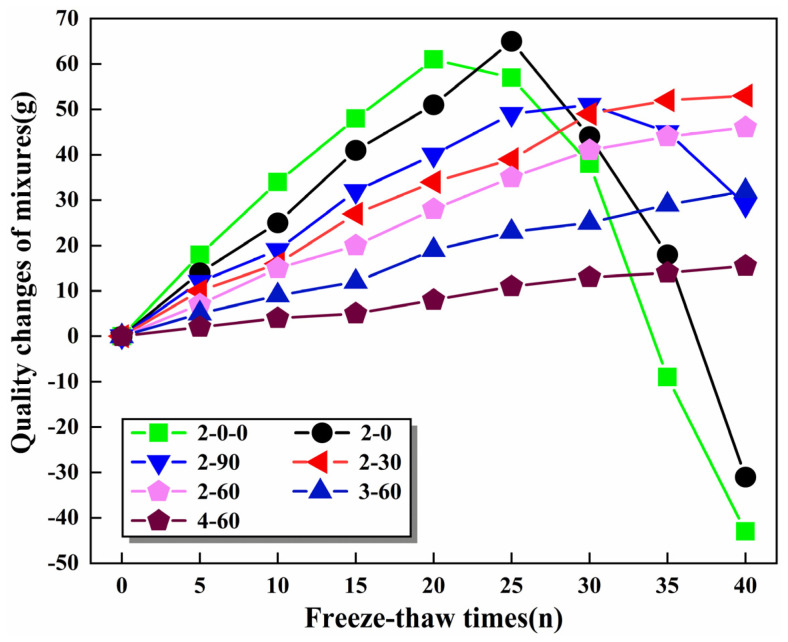
Quality of the mixtures with various freeze–thaw cycles.

**Figure 20 materials-16-06556-f020:**
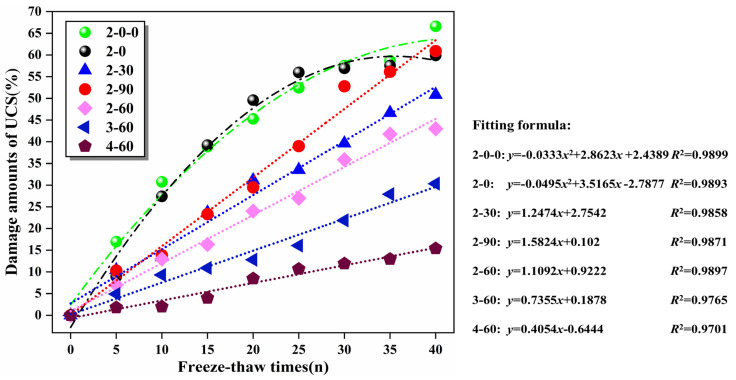
The relationship between UCS damage and freeze–thaw cycles.

**Figure 21 materials-16-06556-f021:**
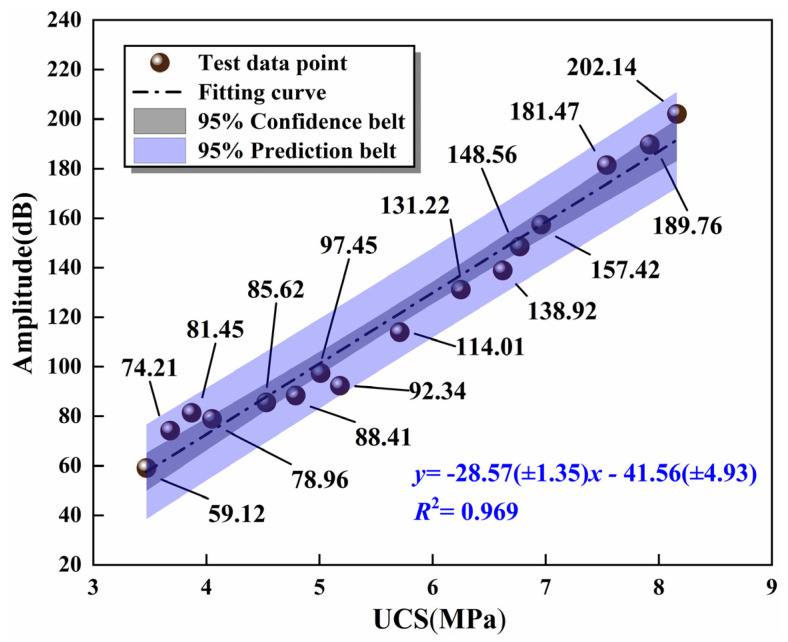
The relationship between amplitude and UCS of the mixtures at the 7 days.

**Figure 22 materials-16-06556-f022:**
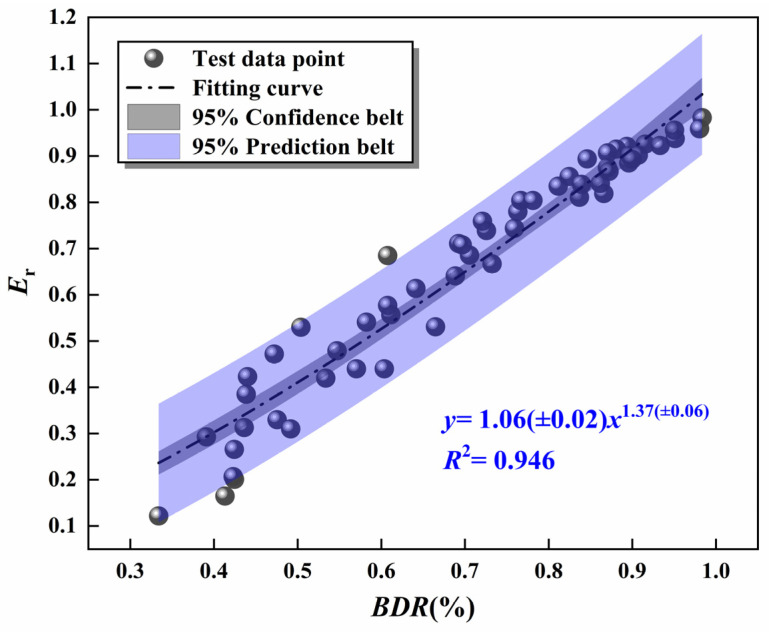
The relationship between *BDR* and *E*_r_ of the mixtures.

**Figure 23 materials-16-06556-f023:**
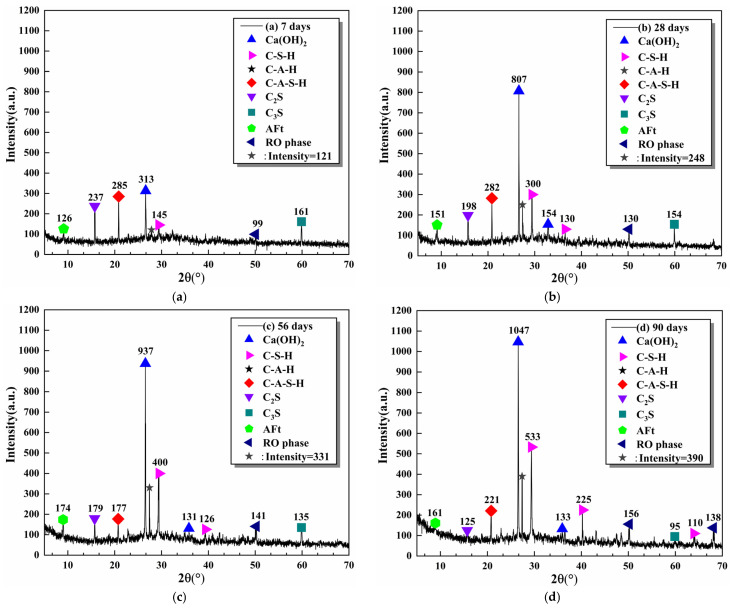
XRD results of the specimen 2–60 at different curing ages: (**a**) 7 days, (**b**) 28 days, (**c**) 56 days, (**d**) 90 days.

**Figure 24 materials-16-06556-f024:**
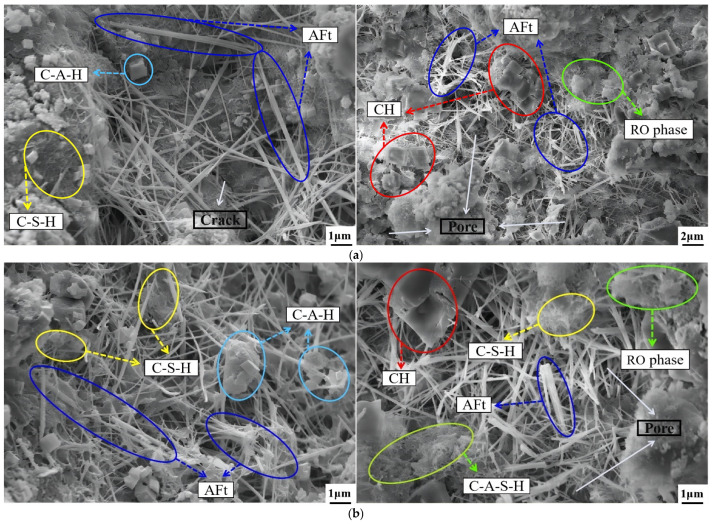
Micro morphology of the specimen 2–60 at different curing ages: (**a**) 7 days, (**b**) 28 days, (**c**) 56 days, (**d**) 90 days.

**Table 1 materials-16-06556-t001:** Chemical compositions of raw materials (%).

Raw Materials	Mass Fraction											
	SiO_2_	Al_2_O_3_	Fe_2_O_3_	CaO	MgO	K_2_O	Na_2_O	TiO_2_	SO_3_	MnO	P_2_O	Trace Substance
SS	15.27	3.21	20.74	48.69	5.52	0.03	0.16	1.94	0.37	-	1.63	2.44
SMS	38.71	6.84	0.96	26.71	11.12	1.44	0.32	0.74	0.81	10.74	-	1.61
RCS	46.94	13.12	6.28	20.01	3.24	1.21	-	-	-	4.45	-	4.75
Cement	17.11	4.14	4.23	64.23	1.52	0.14	0.29	-	2.11	-	-	6.23
FA	45.88	31.56	5.94	4.84	1.31	2.20	0.79	1.61	0.77	-	-	5.10

**Table 2 materials-16-06556-t002:** Main performance indexes of Cement.

Firing Vector (%)	Density (g·cm^−3^)	Specific Surface Area (m^2^·kg^−1^)	Setting Time (min)		Compressive Strength (MPa)		Flexural Strength (MPa)	
			Initial Setting	Final Setting	3 Days	28 Days	3 Days	28 Days
1.68	2.93	394	209	292	29.2	47.3	6.2	7.6

**Table 3 materials-16-06556-t003:** The ratio design of mixtures.

Mixture No	Mass Fraction of Materials (%)					
	RCS (mm)	SS: RCS (mm)		SMS (mm)	FA	Cement
	16~26.5	9.5~16	2.36~9.5	0~2.36		
2–0	19.2	0:15.2	0:26.4	19.2	18	2
2–30	19.2	4.6:10.6	7.9:18.5	19.2	18	2
2–60	19.2	9.2:6	15.8:10.6	19.2	18	2
2–90	19.2	13.8:1.4	23.7:2.7	19.2	18	2
3–0	19.2	0:15.2	0:26.4	19.2	17	3
3–30	19.2	4.6:10.6	7.9:18.5	19.2	17	3
3–60	19.2	9.2:6	15.8:10.6	19.2	17	3
3–90	19.2	13.8:1.4	23.7:2.7	19.2	17	3
4–0	19.2	0:15.2	0:26.4	19.2	16	4
4–30	19.2	4.6:10.6	7.9:18.5	19.2	16	4
4–60	19.2	9.2:6	15.8:10.6	19.2	16	4
4–90	19.2	13.8:1.4	23.7:2.7	19.2	16	4
5–0	19.2	0:15.2	0:26.4	19.2	15	5
5–30	19.2	4.6:10.6	7.9:18.5	19.2	15	5
5–60	19.2	9.2:6	15.8:10.6	19.2	15	5
5–90	19.2	13.8:1.4	23.7:2.7	19.2	15	5

Note: taking 2–60 as an example, 2 means that the cement content in the mixture is 2%, 60 means that the SS of 2.36–9.5 mm and 9.5–16 mm replaces the corresponding particle size RCS with the replacement rate of 60%, and so on.

## Data Availability

The data supporting the findings of this study are available from the corresponding author upon reasonable request.

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
