# Peer review of "Investigating on the Pavement Performance of Multi-Source Solid Wastes by Cement and Fly Ash"

_materials, 2023, doi:10.3390/ma16196556_

Round 1

Reviewer 1 Report

1- line 27: what are C-S-H and AFt?

2- line 58: reference 16 is Mohammadinia et al. [16]

3-Fly ash particles are finer than cement. Thus, the figure 2 seems not correct.

4-In section 3.1, please explain more the reasons for the reduction of OMC by adding cement.

5- What is the application of figure 13? How can these mathematical models be useful?

6- line 465: use Zhao et al. [44] instead of reference [44]. Correct similar errors in the manuscript too.

7-line 192-193: explain “To ensure adequate flow of cold gas, 20 mm gap at least should be reserved around each specimen”

8- line 248: what do you mean by word “significant”? “specific surface area of cement was more significant than FA”

9-lines 249-250: “the water absorption of the specimens was enhanced with the increase of cement content, thereby improving the OMC” but the blue columns in fig.10 show that OMC has been reduced not improved.

10- The UCS values of samples are too large. For example, for cement=5% after 90 days, the UCS reaches 18MPa (18000 kPa). Is it logical? Was the device capable of tolerating this big load without being destroyed?

11- Fig.25 (a) and (b): they seem ettringite not CSH

12- line 139: what are C2S and C3S?

13- fig.22: How did you find amplitude? The ultrasonic device gives velocity (V). explain

The English should be improved. For example: line 32: use “were” instead of “was”. Check the manuscript for other possible mistakes.

Reviewer 2 Report

Dear Authors,

Thank you for the opportunity to review this article.

The topic is timely and necessary, but I'm not sure how innovative the approach is. You write a lot of resources and information yourself and state that this is an under-researched topic. I'm not sure about that.

The title is long, although I understand that it captures the essence.

The abstract is too long and does not meet the requirements of the journal.

The literary state-of-art is not long enough, it needs to be expanded in the context of sustainable construction and the use of similar mixtures in general.

For example about:

10.3390/buildings13051313

10.1016/j.conbuildmat.2023.130791

The description of basic raw materials and materials is quite extensive and well prepared. The description of the test methods is ok, some pictures need to be enlarged, they are unreadable.

Graphs 11 and 12 have a very complicated format.

Other results are beneficial for both science and practice.

Reviewer 3 Report

First of all, the font size in the figures is too small. it can not be read. please increase the size of that.

In Table 1, the sum of the chemical compositions of each material is not 100%. please check it.

In Table 2, what is the mixture proportions of that for evaluation of main performance? please describe it in the paragraph.

In Figure 3, is there any reason you showed only SS and SMS morphology?

In Line 139~ and Figure 4, please explain the process of pre-treatment for measuring XRD patterns. To analyze XRD patterns for each material, did you make them as a powder?

In Line 144, please change from 'mix proportion' to 'mixture proportion'.

In Lines 150-152, I can't understand what you write. what is the meaning that 'the proportion should be between 1:3 and 1:5'?

In Figure 5, it is particle size distribution. and the scale of the x-axis is not a logarithms scale. is it? if not, please change it.

In Table 3, 'SS:RCS' is not clarified. It is difficult to understand this table. is there any reason to show mass fraction?

Figure 6 is not good for understanding the process. please revise it.

This paper doesn't include discussion sections. please add the section to discuss the results. if not, it is just a report.

English writing of the manuscript needs careful checking and revision. 

Kindly consider revising the monotonous style of writing present in this paper.

Reviewer 4 Report

Dear authors,

The topic approached in your manuscript is very interesting from the point of view of the mechanical and frost resistance properties of some pavement materials based on multi-source wastes stabilized by cement and fly ash.

This topic is very well related with the topics of Materials journal.

The authors must clarify some details and data, which are missing. In order to improve this manuscript, I recommend some major changes and improvement as it is shown below.

1. The goal and the main objectives of the research should be written more clearly in the end of the Introduction section.

2. In section 2.1. Raw Materials”, the authors must clarify if the absorption rate (expressed in %) recorded for each raw material, was measured at saturation or after a time of immersion. What is the time of immersion? What standard was used for this measuring?

3. What does the crushing value represent for the raw materials (these are given in section 2.1)?

4. The authors must give more details about all equipment used: laser particle size detector, X-ray fluorescence analyser, SEM equipment, compaction equipment, equipment for UCS and ITS test, equipment for freeze-thaw test, equipment for ultrasonic tests and so on. The details includes: the main technical parameters, model type, manufacturer.

5. Figure 8 contains 3 sub-figures and just 2 titles for sub-figures. The authors must explain the content for all sub-figures.

6. Please write 0% in the following sentence: “The replacement rates of SS were 0, 30%, 60% and 90%, respectively.”

7. In section 2.3.1, it is better that the compaction degree to be expressed in percentage.

8. The authors have to describe the freeze-thaw process. What is the temperature and the time period for freezing? What is the temperature and the time for thawing? How was controlled the depth of the frost? How did the authors control if the freezing was completed fo the entire specimens?

9. The authors must give details about the number of specimens tested for each type of materials.

10. What standard did the authors use for the ultrasonic test?

11. In Figure 10, the authors should show the upper and the lower limits for the results because just the average values are indicated.

12. It could be better if the authors would show the dimensions (length) of the cracks for the various freeze-thaw times in Figure 19.

Dear authors,

Please revise carefully the text of your manuscript and improve the expression in English. 

Round 2

Reviewer 2 Report

I greatly appreciate your active change to the article. But most of the literature is poorly cited.

For example, quotations No. 9 and 10 have a confusion of surnames and first names.

Correct:

Lehner, P.; Hrabová K.

Evaluation of degradation and mechanical parameters and sustainability indicators of zeolite concretes,

Construction and Building Materials, Volume 371, 2023, 130791, ISSN 0950-0618,

https://doi.org/10.1016/j.conbuildmat.2023.130791.

Mostafaei, H.; Badarloo, B.; Chamasemani, N.F.; Rostampour, M.A.; Lehner, P. Investigating the Effects of Concrete Mix Design on the Environmental Impacts of Reinforced Concrete Structures. Buildings 2023, 13, 1313. https://doi.org/10.3390/buildings13051313

Reviewer 3 Report

I checked the revision based on my comments. 

Author Response

Thank you very much for the valuable advice and affirmation to our paper, we wish you a happy life.